# Kinetics of Biomarkers of Oxidative Stress in Septic Shock: A Pilot Study

**DOI:** 10.3390/antiox11040640

**Published:** 2022-03-26

**Authors:** Martin Helan, Jan Malaska, Josef Tomandl, Jiri Jarkovsky, Katerina Helanova, Klara Benesova, Michal Sitina, Milan Dastych, Tomas Ondrus, Monika Pavkova Goldbergova, Roman Gal, Petr Lokaj, Marie Tomandlova, Jiri Parenica

**Affiliations:** 1Department of Anaesthesiology and Intensive Care, St. Anne’s University Hospital Brno, 656 91 Brno, Czech Republic; helan.martin@seznam.cz (M.H.); michal.sitina@fnusa.cz (M.S.); 2International Clinical Research Center (ICRC), St. Anne’s University Hospital Brno, 656 91 Brno, Czech Republic; 3Faculty of Medicine, Masaryk University, 625 00 Brno, Czech Republic; jan.malaska@gmail.com (J.M.); trekar@seznam.cz (K.H.); tomas.ondrus@gmail.cz (T.O.); gal.roman@fnbrno.cz (R.G.); alveus@seznam.cz (P.L.); jiri.parenica@atlas.cz (J.P.); 4Department of Anaesthesiology and Intensive Care Medicine, University Hospital Brno, 625 00 Brno, Czech Republic; 5Department of Biochemistry, Faculty of Medicine, Masaryk University, 625 00 Brno, Czech Republic; tomandl@med.muni.cz; 6Institute of Biostatistics and Analyses, Faculty of Medicine, Masaryk University, 625 00 Brno, Czech Republic; jarkovsky@iba.muni.cz (J.J.); benesova@iba.muni.cz (K.B.); 7Department of Internal Medicine and Cardiology, University Hospital Brno, 625 00 Brno, Czech Republic; 8Department of Laboratory Methods, Faculty of Medicine, Masaryk University, 625 00 Brno, Czech Republic; 35352@mail.muni.cz; 9Department of Pathological Physiology, Faculty of Medicine, Masaryk University, 625 00 Brno, Czech Republic; goldberg@med.muni.cz

**Keywords:** sepsis, septic shock, oxidative stress, antioxidant, biomarker, soluble endoglin, superoxide dismutase, asymmetric dimethylarginine, neopterin

## Abstract

Septic shock is a major cause of mortality in ICU patients, its pathophysiology is complex and not properly understood. Oxidative stress seems to be one of the most important mechanisms of shock progression to multiple organ failure. In the present pilot study, we have analysed eight oxidative-stress-related biomarkers in seven consecutive time points (i.e., the first seven days) in 21 septic shock patients admitted to the ICU. Our objective was to describe the kinetics of four biomarkers related to pro-oxidative processes (nitrite/nitrate, malondialdehyde, 8-oxo-2′-deoxyguanosine, soluble endoglin) compared to four biomarkers of antioxidant processes (the ferric reducing ability of plasma, superoxide dismutase, asymmetric dimethylarginine, mid-regional pro-adrenomedullin) and four inflammatory biomarkers (CRP, IL-6, IL-10 and neopterin). Furthermore, we analysed each biomarker’s ability to predict mortality at the time of admission and 12 h after admission. Although a small number of study subjects were recruited, we have identified four promising molecules for further investigation: soluble endoglin, superoxide dismutase, asymmetric dimethylarginine and neopterin.

## 1. Introduction

Sepsis and especially septic shock are major causes of mortality among ICU patients worldwide, with hospital mortality exceeding 40% [1]. Pathophysiological mechanisms which lead to the development and progression of shock in septic patients and ultimately can lead to organ dysfunction are very complex and still not properly understood. Among all of these mechanisms, oxidative stress seems to play a pivotal role. In normal homeostasis, pro-oxidant and antioxidant processes are tightly regulated to maintain a delicate redox balance. If an infection occurs under physiological circumstances, the oxidative burst of activated innate immune system cells (neutrophils and monocytes) is crucial for host defence. In sepsis, however, oxidative stress results from a dysregulated production of reactive oxygen species (ROS) (superoxide (O2•-), hydrogen peroxide (H2O2), and hydroxyl radical (HO•)) and reactive nitrogen species (RNS) (nitric oxide (NO•) and peroxynitrite (ONOO-)) [2,3]. An increased level of oxidative stress in septic shock has the following pathological effects in particular: lipid peroxidation in cellular membranes, nucleic acid damage and enhanced endothelial dysfunction, which in turn leads to organ damage and dysfunction.

ROS and RNS are very unstable molecules and their direct analysis is, therefore, difficult. However, there are stable biomarkers that are modified by interactions with ROS and RNS and, thus, can be used to measure the extent of oxidative stress. Some of them have already been described as molecules with prognostic potential for several conditions [4,5], including sepsis [6,7].

The main objective of the present study was to explore the kinetics of several oxidative-stress-related biomarkers in seven consecutive time points of septic shock, and thus to identify the most interesting biomarkers for further clinical investigation. With the intention to study pro-oxidant and antioxidant processes separately, we chose substances related to the two opposing mechanisms: (a) biomarkers that show the level of oxidative stress and subsequent tissue damage: nitrite/nitrate (NOx), malondialdehyde (MDA), 8-oxo-2’-deoxyguanosine (8-oxo-dG) and soluble endoglin (sEng); (b) biomarkers of antioxidant mechanisms: the ferric reducing ability of plasma (FRAP), Cu/Zn-superoxide dismutase (SOD), asymmetric dimethylarginine (ADMA) and mid-regional pro-adrenomedullin (MR-proADM). Aiming to compare the extent of oxidative stress with the extent of inflammation, we also decided to analyse inflammatory biomarkers (c), namely C-reactive protein (CRP), neopterin, pro-inflammatory cytokine interleukin 6 (IL-6) and anti-inflammatory cytokine interleukin 10 (IL-10).

## 2. Materials and Methods

### 2.1. Ethical Approval of the Study Protocol

Written informed consent forms were obtained from the subjects either upon their admission to the ICU or after regaining consciousness. As for patients who failed to regain consciousness, anonymous data were processed with the consent of a relative. The study protocol complied with the Declaration of Helsinki [8] and was approved by the local Ethics Committee of the University Hospital Brno (Brno, Czech Republic; no. 02-221014/EK).

### 2.2. Patient Recruitment

The presented study included patients who were admitted to either the Intensive Care Unit (ICU) of the Department of Anaesthesiology and Intensive Care or the Coronary Care Unit (CCU) of the Department of Internal Medicine and Cardiology at the University Hospital Brno in 2016 and who developed septic shock. The exclusion criteria were as follows: malignancy, chronic inflammatory disease or connective tissue disease, trauma, burns and/or pregnancy during the previous three months. Septic shock was defined as hypotension with systolic blood pressure ≤90 mmHg, despite adequate volume resuscitation in patients with proven sepsis, and after the exclusion of other reasons for hypotension. Informed consent was obtained from all subjects involved in the study. In case of impaired consciousness in the time of recruitment the consent was signed by a relative.

### 2.3. Sample Collection

Samples of central venous blood were drawn at seven time points: T1—immediately upon hospital admission, T2—12 h after hospital admission, T3—24 h after hospital admission, T4—morning of the third day (approximately 48 h after hospital admission), T5—morning of the fourth day, T6—morning of the fifth day, and T7—morning of the seventh day.

### 2.4. Laboratory Methods

Immediately after collection, blood samples were centrifuged at 4 °C to obtain plasma, which was subsequently stored at −80 °C. Both the centrifuge and the freezer were located directly at the ICU/CCU. Standard biochemical and haematological blood tests were done immediately after admission. Other laboratory analyses were done within 3–6 months. Plasma concentrations of the compounds of interest were determined as follows: plasma NO levels were evaluated by measuring intermediate products and end products—i.e., nitrite/nitrate (NOx)—that were measured using a colorimetric assay kit based on the Griess test (R&D Systems; detection limit 0.25 µmol/L, inter-assay CV < 5%; reference values—as provided by the manufacturer—range from 10 to 92 µmol/L). Plasma levels of MDA were assessed by HPLC with fluorescent detection (Shimadzu 10A series HPLC system, Shimadzu Corp., Kyoto, Japan) after derivation by thiobarbituric acid (detection limit 0.01 µmol/L, inter-assay CV < 4.2%, and expected values in healthy persons ranging from 0.40 to 1.00 µmol/L), as described by Khoschsorur et al. [9]. Serum concentrations of 8-oxo-dG were measured with ELISA (Cayman Chemicals, MI, USA; detection limit 33 pg/mL, inter-assay CV ˂12%), samples were 51-fold diluted; the kit is highly specific for 8-oxo-dG, whereas 8-oxo-guanosine and 8-oxo-guanine react partly. Expected values in healthy persons range from 0.1 to 0.3 ng/mL [10]. Concentrations of sEng were measured with ELISA (R&D Systems, MN, USA; detection limit 0.007 ng/mL, intra-assay CV < 3.2%, inter-assay CV < 6.7%); expected values in healthy persons range between 2.99 and 7.14 ng/mL.

FRAP was measured using the modified method of Benzie and Strain [11], using a 96-well microplate at 600 nm on a Spectramax 340PC Microplate Reader (Molecular Devices, Silicon Valley, CA, USA). Antioxidant power was calculated using ascorbic acid as the standard (detection limit 10 µmol/L, inter-assay CV ˂ 2.9%); expected values in healthy persons ~1000 μmol/L. Plasma levels of SOD were measured using a sandwich ELISA kit (Bender MedSystems, Vienna, Austria; detection limit 0.040 ng/mL, inter-assay CV < 6%; reference values—as provided by the manufacturer—range from non-detectable to 59.7 ng/mL). Plasma ADMA levels were measured in duplicate by commercially available ELISA kits (DLD Diagnostika, Germany; detection limit 0.05 mmol/L, inter-assay CV < 10%; reference values—as provided by the manufacturer—range from 0.4 to 0.75 µmol/L). Plasma concentrations of MR-proADM were measured using immunofluorescent assay KRYPTOR, B.R.A.H.M.S. (Thermo Fisher Scientific, Germany; detection limit 0.05 nmol/L, inter-assay CV < 6%; reference values—as provided by the manufacturer—< 0.52 nmol/L). Serum concentrations of neopterin were measured using a competitive ELISA kit (IBL International, Hamburg, Germany; detection limit 0.7 nmol/L, inter-assay CV < 10% and cut-off value 10 nmol/L). CRP concentrations were measured using an immunoturbidimetric method (CRPL3 kit; Roche, Basel, Switzerland) on a Cobas 8000 system (detection limit 0.3 mg/L, inter-assay CV 1.99% and cut-off value 5.0 mg/L). For quantitative detection of IL-6 and IL-10 levels, Human Th1/Th2 11plex Kit (Bender MedSystems GmbH, Vienna, Austria) was used, according to manufacturer’s instructions. Concentrations were proportional to fluorescence measured on a flow cytometric system FACSArray (BD Biosciences, San Jose, CA, USA). Data were analysed using the software FlowCytomix™ Pro 2.4 (BD Biosciences, San Jose, CA, USA).

### 2.5. Statistical Analysis

Standard descriptive statistics were used for the analysis; continuous parameters were described by median values supplemented with 5th and 95th percentiles, whereas the occurrence of categorical parameters was described by their count and percentages. Statistical significance of differences between two groups was assessed using the Mann–Whitney test for continuous parameters and the Fisher’s exact test for categorical parameters. Area under ROC curve (AUC) was calculated for each biomarker to determine if the biomarker is a good predictor of the 3-month mortality. The level of statistical significance was set at *p* = 0.05. An IBM SPSS Statistics 28.0.0.0 (IBM Corporation 2021) was used for data analysis.

## 3. Results

A total of 21 patients were enrolled in this pilot study; all of them were mechanically ventilated and the average SOFA score at admission was 12 (8; 17). Twelve patients survived, whereas the remaining nine patients died from multiple organ failure (MOF). There were no statistical differences in the patients’ past medical history, chronic medication or sepsis severity at the time of admission between survivors and non-survivors, except a higher occurrence of lower extremity peripheral artery disease in the group of deceased patients (Table 1).

The kinetics of four substances related to oxidative stress (NOx, MDA, 8-oxo-dG and sEng)—separately for surviving and deceased patients—are shown on the left side of Figure 1a–d. Levels of NOx (Figure 1a) were highest at the time of admission (T1) and then decreased continuously to the lowest levels on the seventh day (T7). There was a statistically non-significant trend to the higher admission NOx levels in surviving patients than in those who died eventually. There were no marked changes in MDA levels (Figure 1b) over time. Levels of 8-oxo-dG (Figure 1c) decreased from the time of admission (the highest concentrations were measured on T1) until day 4 (T5) in both groups; after that (T5–T7), changes in concentration were not observed. There were almost no changes in sEng levels in the group of surviving patients, but sEng levels were initially significantly increased in the group of deceased patients and a drop was only observed on the 5th day (Figure 1d).

The analysis of the FRAP showed a downward trend from the time of admission to T4; after that, its value remained approximately the same. There were no differences among the two groups in this regard (Figure 1e). Levels of SOD (Figure 1f) were observed to decrease in the period T1–T4 in the group of surviving patients, whereas they remained significantly elevated over the entire seven-day period in the group of deceased patients. Regarding ADMA levels, significantly higher levels were observed in the group of deceased patients at admission and then a robust increase was observed in the same group in the period T5–T6 (Figure 1g). Admission levels of MR-proADM (Figure 1h) had a nonsignificant trend to be higher in the group of deceased patients; in both groups, a consistent downward trend was observed until T7.

Figure 1i shows the changes in the CRP levels, i.e., the highest values during the first 3 days and a decrease afterwards. The elevation of CRP levels persisted for a longer time in the group of deceased patients. Neopterin levels had a trend of being increased in the group of deceased patients (with a maximum on T4) (Figure 1j). Finally, interleukins 6 and 10 demonstrated a robust increase, especially in the deceased patients with high interindividual variability.

Table 2 compares median admission levels of biomarkers in surviving and deceased patients, together with the biomarkers’ ability to predict the 3-month mortality, expressed by the AUC of the C-statistic. According to the C-statistic, sEng (*p* = 0.002; AUC 0.976), ADMA (*p* = 0.040; AUC 0.769) and SOD (*p* = 0.039; AUC 0.773) were the best predictors of mortality risk at admission. Twelve hours after admission, sEng (*p* = 0.010, AUC 0.958) and neopterin (*p* = 0.013, AUC 0.875) were significant predictors of mortality (Table 3). Elevated sEng levels over the first five days (>4 ug/L) are markers of a poor prognosis. Changes of the values after the first 12 h after admission (T1–T2; data not shown) in any of the biomarkers does not significantly predict mortality.

## 4. Discussion

Patients were admitted to the ICU with septic shock and with systemic inflammatory response syndrome (SIRS) already having been developed, as indicated by the high admission levels of the CRP. The maximum CRP levels were observed approximately 12–24 h after admission. Based on the known kinetics of the CRP (with a half-life of 24–48 h), it can be assumed that the primary inflammatory response was set off about 12–24 h before admission. Then, 24–48 h after a peak in CRP levels, a decrease was observed, which corresponds to a response to the initiation of a comprehensive treatment of septic shock. In the group of deceased patients, an increase in inflammatory parameters was observed on the 4th–5th day. This flare-up of the inflammatory reaction was even more marked in neopterin levels. Neopterin is produced by activated macrophages and is a marker of cellular immunity activation [12]. At the same time, however, neopterin concentrations allow one to estimate the extent of oxidative stress elicited by the immune system [13]. Neopterin becomes significantly increased in deceased patients 12 h after admission, while at the time of admission the difference is not yet apparent. IL-6 and IL-10 were both robustly deregulated just from the beginning of septic shock; findings even more pronounced in deceased patients. This corresponds with a so-called cytokine storm.

The increased production and activity of inducible nitric oxide synthase (iNOS), which is responsible for the dysregulated overproduction of NO, has been described in the pathophysiology of septic shock. High levels of NO contributed to an even deeper state of shock by increasing vasodilation, capillary permeability and cardiodepressant effects, and led to an increased production of toxic reactive nitrogen species [14]. In our cohort, the highest levels of NOx were observed at admission and decreased afterwards. NOx production had a trend of being increased (*p* = 0.073) in the group of surviving patients (when compared to the group of deceased patients). This observation contrasts with findings published in several previous studies, which described higher levels of NOx in deceased patients than in survivors [15,16,17]. This contradiction can be potentially explained by a high interindividual variability of NOx levels in combination with the small number of patients included in our study [16]. On the other hand, it can be speculated that even an increase in NO (to some degree) may be beneficial under certain conditions. For example, some studies have shown that NO inhibitor administration has increased patient mortality [17]. However, in general, it should be noted that the results of pharmacological experiments (using both inhibitors or donors) to influence the level and effect of NO in patients with sepsis are inconsistent.

Apart from NOx, we have also observed an increased production of 8-oxo-dG, which reflects the extent of oxidative damage to nucleic acids. The maximum levels of 8-oxo-dG were measured at admission and decreased from the second day onwards. Elevated MDA levels represent another marker of cell damage caused by oxidative stress. MDA is the end product of lipid peroxidation, lipids being the key component of cell membranes. MDA production was more accentuated in the group of deceased patients, but this difference was not statistically significant in the first 24 h, and therefore, is not a suitable predictor for the patients’ prognosis. The dysregulated activation of endothelial cells is another key mechanism leading to the development of septic shock, resulting in an increased permeability and dysfunction of the endothelial barrier [18]. Endoglin is part of the transmembrane TGF-β receptor complex located on the surface of endothelial cells. The extracellular domain of membrane-bound endoglin can be proteolytically cleaved, leading to the production of sEng. sEng production is activated by oxidative stress [19,20,21,22] and can be considered to be a marker of endothelial activation/dysfunction [23]. Endoglin is also expressed on monocytes, where it has a regulatory function in the immune response [24], and contributes to the regulation of NO production [25]. Markedly higher levels of sEng in the group of deceased patients over the entire seven-day follow-up period might reflect a high degree of endoglin production induced by oxidative stress and, therefore, a high degree of endothelial dysfunction. From the clinical point of view, it is important to note that sEng levels are significantly different over the first four days (Figure 1d, Table 2 and Table 3).

The overall antioxidant capacity of an individual is expressed by the FRAP. The FRAP depends on the present concentration of molecules with antioxidant properties (uric acid, antioxidant vitamins C [26] and E, proteins, bilirubin and other compounds). The maximum antioxidant capacity was measured at admission and then was observed to decrease, regardless of the patients’ prognosis. The antioxidant effect of SOD is based on the conversion of extremely toxic and reactive superoxide radicals to less toxic hydrogen peroxide. Again, the SOD concentration was at its highest level at admission and decreased over time in the group of surviving patients (most likely reflecting the decrease in oxidative stress as a result of successful treatment); this contrasted with findings in the group of deceased patients, where the SOD concentration was observed to decrease initially but then rise again. The AUC values correspond to this observation. SOD might be a good predictor of a poor outcome immediately upon admission (AUC = 0.773), but later (after 12 h) its potential decreases significantly. This fact certainly reduces the possibilities of clinical use. ADMA competes with L-arginine for the active site of NOS and thus serves as one of the regulatory mechanisms for NO production. The increased production of ADMA in patients with septic shock might cause (among other causes) a decrease in the basal production of NO. This is in accordance with our results. A markedly higher production of ADMA was observed on the 4th–5th day in the group of patients who died eventually. Adrenomedullin protects the cells from damage by oxidative stress [27,28]. The levels of MR-proADM were also highest at admission and were observed to decrease in the group of survivors from the 3rd day onwards; by contrast, MR-proADM levels remained to be high in the group of patients who died eventually, most likely reflecting the ongoing high level of oxidative stress.

Damage to nucleic acids, lipid peroxidation of membranes and increased apoptosis are indicative of organ damage and multiple organ dysfunction syndrome, the severity of which correlates with the mortality of patients in septic shock [29].

Importantly, our work has documented an extraordinary increase in the levels of several oxidative stress biomarkers, particularly at the initial stage of septic shock. These levels then decreased in patients who survived (possibly as a response to the initial treatment); on the other hand, such a marked decrease was not observed in patients who died eventually (see the development of SOD, ADMA, MDA and sEng levels). This finding suggests that sepsis might potentially be treated with targeted antioxidant therapy. The administration of vitamin C, thiamine, acetylcysteine and other substances with an antioxidant effect has been recently tested in pilot studies [30,31,32,33]. Particularly promising outcomes have been described for the therapeutic combination of vitamin C, thiamine and hydrocortisone, suggesting that the early administration of this therapy might shorten the duration of severe sepsis and septic shock, prevent progressive organ dysfunction (including acute kidney injury) and reduce the patients’ mortality [34]. However, it will be necessary to verify this therapeutic effect in larger clinical studies.

The main limitations of our study are the small number of patients enrolled and the monocentric nature of the study. This can underestimate the significance in the results, but it can also cause a small number error. Sepsis is a very heterogeneous syndrome with a number of factors (pathogen, source of infection, patient phenotype, etc.), the individual effects of which we have not been able to verify.

## 5. Conclusions

Based on our analysis of selected oxidative stress biomarkers over the first seven days of septic shock, we have documented an association between a severe inflammatory response and a high level of oxidative stress (NOx), which subsequently leads to lipid peroxidation in cell membranes (MDA), damage to nucleic acids (8-oxo-dG), the development of endothelial dysfunction (sEng), and thus probably to multiple organ failure, too. Antioxidant mechanisms were markedly more intensified in patients who died eventually (SOD, MR-proADM, ADMA). The described pathophysiological associations are summarised in Figure 2. Monitoring the kinetics of oxidative stress markers might be useful for the identification of patients with a poor prognosis and for decision making, such as whether to escalate treatment aggressiveness. Especially, soluble endoglin, superoxide dismutase, asymmetric dimethylarginine and neopterin seem be the most suitable biomarkers for further clinical investigation.

## Figures and Tables

**Figure 1 antioxidants-11-00640-f001:**
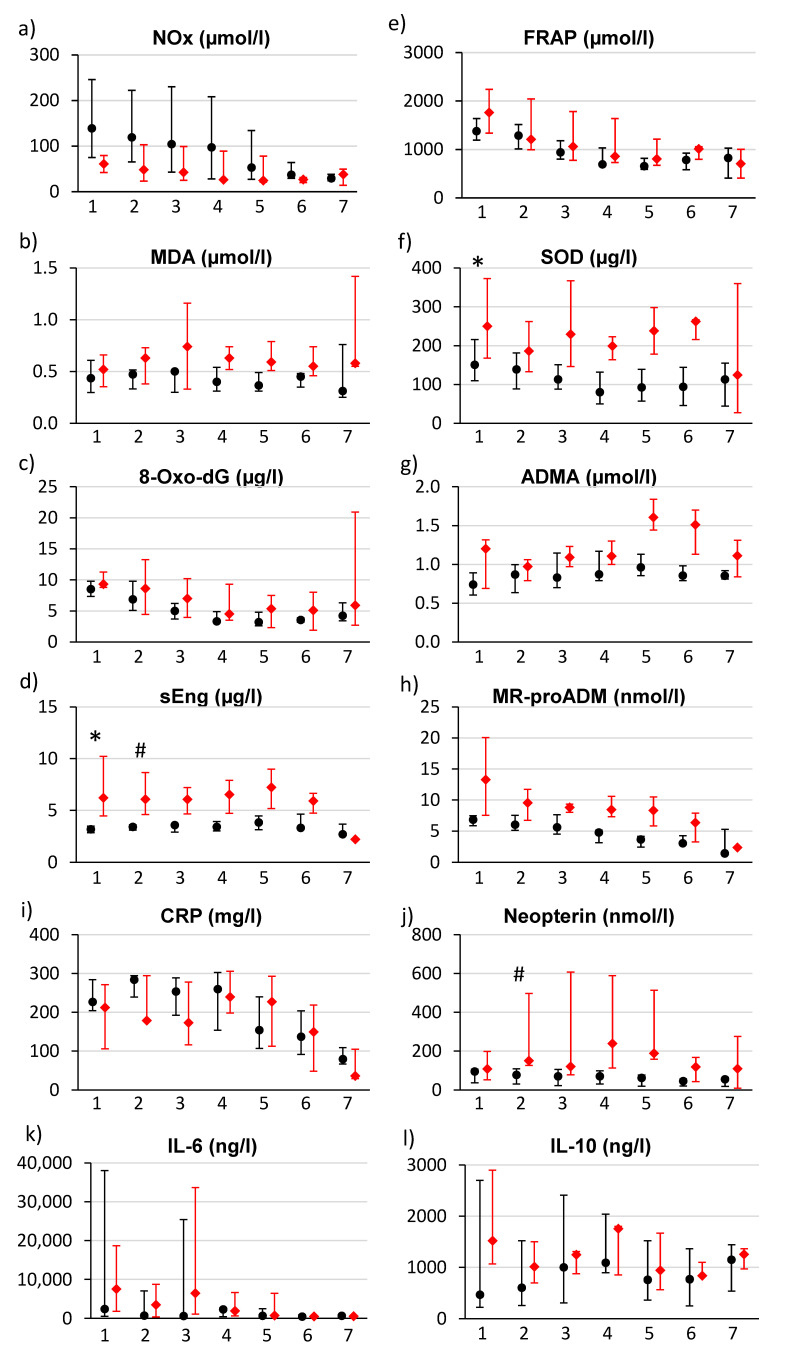
Course of biomarker changes in surviving patients (black points) and in deceased patients (red squares). Seven samples from each timepoint T1–T7 (T1—immediately upon hospital admission, T2—12 h after hospital admission, T3—24 h after hospital admission, T4—morning of the third day (approximately 48 h after hospital admission), T5—morning of the fourth day, T6—morning of the fifth day, T7—morning of the seventh day). Median and interquartile ranges are presented for each time point. * indicates statistical significance at T1, # indicates statistical significance at T2.

**Figure 2 antioxidants-11-00640-f002:**
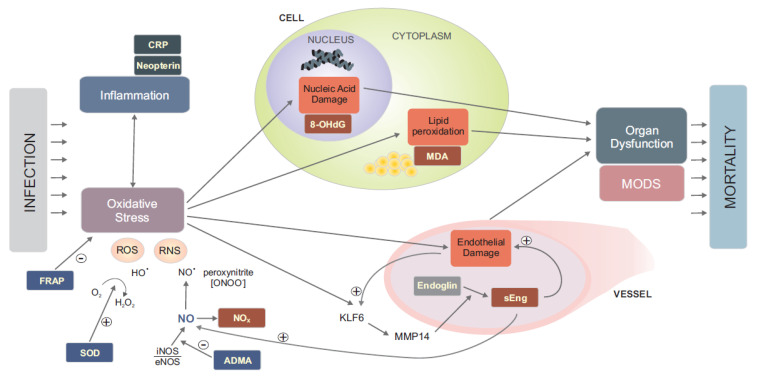
The diagram shows the presumed context of pathophysiological processes with sepsis. With sepsis, the host response to infection leads to deregulation of the inflammatory response. This is characterised by elevated CRP and neopterin levels. At the same time, oxidative stress is deregulated by increased ROS and RNS production. An indicator of excessive NO production is an increased NOx level. Opposite antioxidation mechanisms are also activated: SOD catalyses the conversion of the superoxide radical to the less toxic and reactive hydrogen peroxide and ADMA competitively suppresses NO synthesis. The level of plasma antioxidant capacity can be determined as FRAP. Oxidative stress leads to cell damage by lipid peroxidation and nucleic acid damage. The level of these phenomena is described by MDA and 8-OHdG markers, respectively. Endothelial dysfunction also develops. Through the activation of the transcription factor KLF6 and subsequent MMP14 metalloproteinase, a massive release of soluble endoglin from the endothelium occurs. Its function in respect of sepsis has not yet been clearly identified. It is not clear whether it is only due to or causes endothelial dysfunction. Among other things, it increases the production of NO. Damage to membranes, nucleic acids and the endothelium, caused by oxidative stress, leads to tissue damage, subsequent organ dysfunction and the development of multiorgan failure. The level of organ dysfunction correlates to the mortality of patients who are in septic shock.

**Table 1 antioxidants-11-00640-t001:** Baseline characteristics of the study population, comparison of surviving and deceased septic shock patients.

	Total (*n* = 21)	Surviving (*n* = 12)	Deceased (*n* = 9)	*p*
Basic characteristics				
Male	15 (71.4%)	8 (66.7%)	7 (77.8%)	0.659
Female	6 (28.6%)	4 (33.3%)	2 (22.2%)	0.659
Age (years)	65 (26; 76)	64 (33; 76)	65 (26; 75)	0.972
BMI (kg/m^2^)	29 (23; 51)	30 (24; 51)	29 (23; 36)	0.602
Systolic BP (mmHg)	100 (40; 150)	100 (40; 145)	100 (70; 150)	0.277
Diastolic BP (mmHg)	50 (30; 80)	50 (30; 60)	60 (50; 80)	0.018
Past medical history				
Smoking	6 (28.6%)	2 (16.7%)	4 (44.4%)	0.331
Hypertension	15 (71.4%)	9 (75.0%)	6 (66.7%)	0.999
Diabetes mellitus	4 (19.0%)	2 (16.7%)	2 (22.2%)	0.999
Hyperlipoproteinaemia	8 (38.1%)	4 (33.3%)	4 (44.4%)	0.673
Previous CAD	2 (9.5%)	1 (8.3%)	1 (11.1%)	0.999
History of stroke/TIA	3 (14.3%)	2 (16.7%)	1 (11.1%)	0.999
COPD	2 (9.5%)	0 (0.0%)	2 (22.2%)	0.171
Atrial fibrillation	3 (14.3%)	2 (16.7%)	1 (11.1%)	0.999
Medication at admission				
Antiplatelet drugs	5 (23.8%)	2 (16.7%)	3 (33.3%)	0.611
Anticoagulants	1 (4.8%)	0 (0.0%)	1 (11.1%)	0.429
ACE inhibitors	2 (9.5%)	0 (0.0%)	2 (22.2%)	0.171
Beta-blockers	7 (33.3%)	5 (41.7%)	2 (22.2%)	0.642
Statins	4 (19.0%)	2 (16.7%)	2 (22.2%)	0.999
ARBs	4 (19.0%)	3 (25.0%)	1 (11.1%)	0.603
Diuretics	2 (9.5%)	2 (16.7%)	0 (0.0%)	0.486
Spironolactone	2 (9.5%)	2 (16.7%)	0 (0.0%)	0.486
Ca-blockers	2 (9.5%)	1 (8.3%)	1 (11.1%)	0.999
Oral antidiabetic drugs	3 (14.3%)	2 (16.7%)	1 (11.1%)	0.999
Sepsis severity, organ dysfunction				
APACHE II	28 (19; 42)	28 (18; 37)	30 (20; 46)	0.465
SOFA	12 (7; 18)	12 (10; 17)	12 (7; 18)	0.807
Lactate (mmol/L)	3.4 (0.9; 12.7)	3.3 (0.9; 8.4)	3.9 (1.3; 12.7)	0.499
Creatinin (µmol/L)	153 (39; 570)	146 (39; 391)	196 (52; 570)	0.508
Trombocytes (10^9^/L)	131 (26; 231)	112 (51; 231)	150 (26; 190)	0.761
Source of sepsis				
Pneumonia	8 (38.1%)	4 (33.3%)	4 (44.4%)	0.673
Abdominal infection	7 (33.3%)	6 (50.0%)	1 (11.1%)	0.159
Urosepsis	1 (4.8%)	0 (0.0%)	1 (11.1%)	0.429
CRBI	1 (4.8%)	0 (0.0%)	1 (11.1%)	0.429
Meningitis	2 (9.5%)	1 (8.3%)	1 (11.1%)	1.000
Multiple or unknown	2 (9.5%)	1 (8.3%)	1 (11.1%)	1.000

Median (minimum; maximum) values are presented for continuous variables, absolute and relative frequencies for binary variables. For the comparison of deceased and surviving patients, *p*-values of Mann–Whitney U test are presented for continuous variables and *p*-values of Fisher’s exact test are presented for binary variables. ACE, angiotensin-converting enzyme; ARBs, angiotensin II receptor blockers; BMI, body mass index; BP, blood pressure; CAD, coronary artery disease; COPD, chronic obstructive pulmonary disease; CRBI, catheter-related bloodstream infection; SOFA, sequential organ failure assessment score; PAD, peripheral artery disease; TIA, transient ischaemic attack.

**Table 2 antioxidants-11-00640-t002:** Comparison of oxidative stress biomarkers and clinical characteristics values (at admission —T1) in surviving and deceased patients and their ability to predict 3-month mortality according to C-statistics.

Biomarker	Surviving (*n* = 12)	Deceased (*n* = 9)	*p* Value	Cut-off Value	AUC	Sensitivity	Specificity
NOx (µmol/L)	139.0 (58.1; 267.0)	61.0 (39.7; 161.5)	0.073	≤80.5	0.810	0.833	0.714
MDA (µmol/L)	0.44 (0.26; 0.65)	0.52 (0.33; 0.89)	0.219	≥0.32	0.667	1.000	0.417
8-oxo-dG (µg/L)	8.48 (5.63; 19.47)	9.30 (8.06; 22.26)	0.155	≥8.3	0.690	0.889	0.500
sEng (µg/L)	3.17 (2.70; 3.73)	6.21 (3.95; 10.36)	0.002 *	≥3.63	0.976	1.000	0.857
FRAP (µmol/L)	1377 (925; 2130)	1762 (923; 2601)	0.422	≥2 211	0.611	0.333	1.000
SOD (µg/L)	150.5 (82.6; 364.9)	249.9 (141.4; 939.0)	0.039 *	≥134.2	0.773	1.000	0.500
ADMA (µmol/L)	0.74 (0.42; 1.09)	1.20 (0.62; 1.83)	0.040 *	≥1.16	0.769	0.556	1.000
MR-proADM (nmol/L)	6.82 (4.34; 21.45)	17.06 (7.52; 24.72)	0.240	≥7.50	0.722	0.833	0.833
CRP (mg/L)	227.1 (112.6; 316.4)	212.4 (83.4; 327.6)	0.553	≤138.9	0.583	0.444	0.917
Neopterin (nmol/L)	94.9 (12.5; 155.1)	109.2 (33.5; 531.6)	0.305	≥122.0	0.646	0.500	0.833
IL-6 (ng/L)	2445 (398; 191,782)	7617 (837; 64,783)	0.651	≥1616	0.567	0.889	0.500
IL-10 (ng/L)	467 (192; 7356)	1520 (397; 75,390)	0.111	≥645	0.713	0.889	0.667
SOFA	12.0 (9.6; 16.5)	12.0 (7.8; 17.6)	0.807	≤10.5	0.537	0.333	0.833
Lactate (mmol/L)	3.25 (0.90; 7.85)	3.90 (1.50; 10.62)	0.499	≥3.35	0.593	0.667	0.583
Creatinin (µmol/L)	146.0 (57.7; 352.5)	196.0 (58.8; 476.4)	0.508	≥182.0	0.593	0.556	0.750
Trombocytes (10^9^/L)	112.0 (61.7; 229.5)	150.0 (31.6; 292.1)	0.761	≥114.0	0.500	0.750	0.545

Biomarkers and baseline clinical characteristics at time point T1 (at admission). Results are expressed as median values (5th percentile; 95th percentile). *p*-values were calculated by Mann–Whitney U test. The biomarkers’ ability to predict mortality was expressed as AUC of ROC curve. * indicates statistical significance. NOx, nitrite/nitrate; MDA, malondialdehyde; 8-oxo-dG, 8-oxo-2′-deoxyguanosine; sEng, soluble endoglin; FRAP, ferric reducing ability of plasma; SOD, superoxide dismutase; ADMA, asymmetric dimethylarginine; MR-proADM, mid-regional pro-adrenomedullin; CRP, C-reactive protein; IL-6, interleukin 6; IL-10, interleukin 10; SOFA, sequential organ failure assessment score.

**Table 3 antioxidants-11-00640-t003:** Comparison of oxidative stress biomarkers’ values and clinical characteristics (12 h after admission—T2) in surviving and deceased patients and their ability to predict 3-month mortality according to C-statistics.

Biomarker	Surviving (*n* = 12)	Deceased (*n* = 9)	*p*	Cut-Off Value	AUC	Sensitivity	Specificity
NOx (µmol/L)	119.0 (44.7; 237.4)	48.0 (22.9; 148.0)	0.138	≤57.5	0.762	0.667	0.857
MDA (µmol/L)	0.47 (0.29; 0.64)	0.63 (0.35; 1.15)	0.167	≥0.59	0.702	0.571	0.917
8-oxo-dG (µg/L)	6.86 (3.87; 13.27)	8.60 (4.27; 20.92)	0.591	≥8.4	0.583	0.571	0.750
sEng (µg/L)	3.41 (3.08; 3.60)	6.06 (3.83; 9.92)	0.010 *	≥4.10	0.958	0.833	1.000
FRAP (µmol/L)	1288 (822; 1649)	1209 (667; 2118)	0.866	≥1848	0.470	0.286	1.000
SOD (µg/L)	138.6 (72.0; 331.7)	186.0 (128.1; 457.3)	0.120	≥118.8	0.726	1.000	0.417
ADMA (µmol/L)	0.87 (0.57; 1.33)	0.97 (0.78; 1.45)	0.340	≥0.72	0.643	1.000	0.417
MR-proADM (nmol/L)	6.01 (4.46; 13.19)	9.54 (4.02; 15.89)	0.309	≥6.64	0.694	0.833	0.667
CRP (mg/L)	284.2 (176.2; 307.1)	178.8 (133.8; 389.0)	0.237	≤191.9	0.673	0.571	0.917
Neopterin (nmol/L)	77.7 (13.1; 114.6)	151.0 (50.0; 541.3)	0.013 *	≥122.5	0.875	0.833	1.000
IL-6 (ng/L)	733(193; 79,268)	3515 (342; 140,294)	0.432	≥916	0.619	0.714	0.583
IL-10 (ng/L)	606 (114; 2556)	1011 (326; 6938)	0.64	≥521	0.571	0.857	0.500
SOFA	12.0 (8.0; 15.9)	13.0 (6.4; 19.2)	0.887	≥17.5	0.639	0.222	1.000
Lactate (mmol/L)	2.45 (0.90; 7.08)	1.80 (1.49; 8.77)	0.703	≥1.35	0.440	1.000	0.333
Creatinin (µmol/L)	136.0 (53.9; 327.6)	213.0 (76.6; 475.8)	0.261	≥205.5	0.667	0.714	0.833
Trombocytes (10^9^/L)	92.6 (44.1; 231.6)	105.0 (23.2; 213.9)	0.965	≤54.0	0.488	0.375	0.900

Biomarkers and clinical characteristics at time point T2 (12 h after admission). Results are expressed as median values (5th percentile; 95th percentile). *p*-values were calculated by Mann–Whitney U test. The biomarkers’ ability to predict mortality was expressed as AUC of ROC curve. * indicates statistical significance. NOx, nitrite/nitrate; MDA, malondialdehyde; 8-oxo-dG, 8-oxo-2′-deoxyguanosine; sEng, soluble endoglin; FRAP, ferric reducing ability of plasma; SOD, superoxide dismutase; ADMA, asymmetric dimethylarginine; MR-proADM, mid-regional pro-adrenomedullin; CRP, C-reactive protein; IL-6, interleukin 6; IL-10, interleukin 10; SOFA, sequential organ failure assessment score.

## Data Availability

Data is contained within the article.

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
