# Peer review of "Kinetics of Biomarkers of Oxidative Stress in Septic Shock: A Pilot Study"

_antioxidants, 2022, doi:10.3390/antiox11040640_

Round 1

Reviewer 1 Report

The manuscript entitled “Kinetics of biomarkers of oxidative stress in septic shock: a pilot study” by Martin Helan et al, explores the kinetics of several oxidative stress-related biomarkers in seven consecutive time points of septic shock and identified some biomarkers for further clinical investigation. With this they studied pro-oxidant and antioxidant processes separately, they also chose substances related to the two opposing mechanisms.

The authors also aim to compare the extent of oxidative stress with the extent of inflammation and decided to analyze two inflammatory biomarkers (c), namely C-reactive protein (CRP) and neopterin additionally.

Overall, this is an interesting concept, however there are some important questions that needs to be addressed to support this study.

General comments:

  1. In the present study authors have observed 8 oxidative stress markers I consecutive time points in 21 septic shock patients which were admitted to ICU, but I wonder, whether this number of patients can be good enough to draw a conclusion for selecting a biomarker.
  2. In addition, authors have also checked two inflammatory biomarkers (CRP, neopterin), but here again I am not sure whether there could be only these two inflammatory markers de-regulated. I suggest, authors must check some pro-inflammatory and anti-inflammatory markers as well.
  3. Authors should also have some model, say at least some cell culture model to validate these findings.

Reviewer 2 Report

Line 83. Quoted the reference for the Declaration of Helsinki

Line 84 reference to the number of the clinical protocol approved by the University Hospital Brno (Brno, Czech Republic).

Figure 1. Difficult to distinguish between black and gray lines, please change to solid and dashed lines

Reviewer 3 Report

In the present pilot study the authors analysed the role of oxidative stress in septic shock, with the aim of finding potential biomarkers of mortality risk.

I would like to ask the authors a few questions:

1. The authors report that the patients were recruited in 2016, and that the frozen samples were analyzed at 3-6 months. Why has it taken more than 5 years to analyze the results?

2.The number of patients is limited, as the authors acknowledge.  Do the authors think this underpowers the differences between parameters?

3.In Table 1: if the data related to men were included, the total survival rate would be known, otherwise the reader would have to do calculations. Have the authors considered whether gender is relevant in the evolution of sepsis?

4.Figure 1 is very illustrative, and the great dispersion of the values is evident. It would be appropriate to put the statistics, and the legend should include all the information (indicate what the numbers 1 to 7 of the abscissa axis refer to)

5. About the "promising molecules" identified: the SOD ranges at admission and the neopterin ranges after 12 hours of admission are too wide. Do you think this is a problem to be considered a good biomarker of mortality risk?

Round 2

Reviewer 1 Report

Regarding Manuscript ID: antioxidants-1613093.

Authors provided expression for IL6 and IL10.

Authors have answered the other questions and provided excuses or lack of feasibility not to perform other experiments.

Author Response

Thank you for your review.

Reviewer 3 Report

The authors have adequately responded to the reviewers' concerns. They have also expanded the number of data in the article, showing cytokine data, thus increasing the consistency of their observations.

Author Response

Thank you for your review.